# Transcriptome and Metabolome Analysis Reveals the Importance of Amino-Acid Metabolism in *Spodoptera Frugiperda* Exposed to Spinetoram

**DOI:** 10.3390/insects13090852

**Published:** 2022-09-19

**Authors:** Zupeng Gao, Raufa Batool, Weifeng Xie, Xiaodan Huang, Zhenying Wang

**Affiliations:** 1State Key Laboratory for Biology of Plant Diseases and Insect Pests, Institute of Plant Protection, Chinese Academy of Agricultural Sciences, Beijing 100193, China; 2Engineering Research Center of Natural Enemy Insects/Institute of Biological Control, Jilin Agricultural University, Changchun 130118, China

**Keywords:** spinetoram, *spodoptera frugiperda*, metabolism, amino acid, physio-biochemistry

## Abstract

**Simple Summary:**

The spinetoram was high toxicity toward *Spodoptera frugiperda* larvae in, and was recommended to control *Spodoptera frugiperda* in China. In order to explore the effect of spinetoram on *S. frugiperda*, we studied the effect of spinetoram on amino acid metabolism of *S. frugiperda* by transcriptome analysis and LC-MS/MS analysis. The results showed this pesticide induced the change of arginine, glutamic acid, aspartic acid, and lysine cause in larvae of *S. frugiperda*, which was one of the factors of larval death. And, the down-regulation of phenylalanine may retard the tricarboxylic acid cycle to produce GTP. The decrease in citric acid is associated with an increase in leucine. However, enhancement of glucose metabolism and tryptophan provides the basis for the restoration of normal physiological functions of *S. frugiperda* larvae.

**Abstract:**

Pests are inevitably exposed to sublethal and lethal doses in the agroecosystem following the application of pesticides indispensable to protect food sources. The effect of spinetoram on amino-acid metabolism of fall armyworm, *Spodoptera frugiperda* (J.E. Smith), was investigated, at the dose of LC_10_ and LC_90_, by transcriptome and LC-MS/MS analysis. Using statistics-based analysis of both POS and NEG mode, a total of 715,501 metabolites in *S. frugiperda* were significantly changed after spinetoram treatment. The enhancement of glucose metabolism provides energy support for detoxification in larvae. The decrease in valine and isoleucine is associated with an increase in leucine, without maintaining the conservation of citric acid in the larvae. The down-regulation of phenylalanine may retard the tricarboxylic acid cycle to produce GTP. The abundance of lysine was decreased in response to spinetoram exposure, which damages the nervous system of the larvae. The abundance of arginine increases and causes non-functional contraction of the insect’s muscles, causing the larva to expend excess energy. Tryptophan provides an important substrate for eliminating ROS. The changes in glutamic acid, aspartic acid, and lysine cause damage to the nerve centers of the larvae. The results of transcriptome and LC-MS/MS analysis revealed the effects of pesticide exposure on amino-acid metabolism of *S. frugiperda* successfully and provide a new overview of the response of insect physio-biochemistry against pesticides.

## 1. Introduction

The fall armyworm, *Spodoptera frugiperda* (J.E. Smith) (Lepidoptera: Noctuidae), is one of the migratory polyphagous pests of the neotropics of the Americas [1]. Due to the stronger migration ability of the adults, they rapidly spread from America to over 100 countries worldwide including China [2,3]. It has been reported to attack 353 host plant species from 76 plant families, and the economic damage from *S. frugiperda* can reach 73% worldwide from maize alone [4]. In China alone, the estimated damage area loss due to *S. frugiperda* was more than 134.77 million ha in 2020 [5]. Because chemical insecticides have rapid effects, especially during pest outbreak periods, the farmers often use massive amounts of chemical insecticides to control pests. However, the *S. frugiperda* has evolved resistance to various insecticides, and it has been confirmed that the *S. frugiperda* has potential resistance genes to carbamate and organophosphorus insecticides in many provinces of China [6,7]. Spinetoram is a new member of the spinosyn family, and it acts on the insect nervous system at the nicotinic acetylcholine receptor (nAChR) and gamma-aminobutyric acid receptor (GABA) [8]. Due to its high toxicity toward *S. frugiperda* larvae in laboratory bioassays and field applications, spinetoram was recommended to control *S. frugiperda* by the Ministry of Agriculture and Rural Affairs of the People’s Republic of China in 2020 [9].

Insecticides are mainly used by foliar spray and seed coating, so they exert toxic effects via ingestion. When insecticides are influenced by environmental degradation or via heterogeneous spatial coverage on individual plants, the exposed individuals are influenced by the sublethal dose of insecticides [10]. So, the study of sublethal effects is recommended as an emerging tool to assess the environmental risk of insecticides on arthropods. The sublethal dose of insecticides can negatively affect the larval duration and larval weight in *Spodoptera exigua* [11], pupal weight, and pupation rate in *Plutella xylostella* [12], adult longevity and reproduction in *Spodoptera litura* [13]. Several scientists have recently reported that the sublethal effects of pesticide have negative impacts on the visual learning and immunocompetence of honeybees [14] and the embryo development of *Eriopis connexa* [15]. The lethal dose of imidacloprid have negative impacts on aversive learning and movement of honeybees and that chronic exposure effects were dose-dependent for the insecticide [16].

With the development of third-generation sequencing technology, it gives us a chance to comprehensively understand gene expression after pesticides are used to treat the insect. The transcriptome can intuitively observe the changes in insects’ energy metabolism under pesticide stress. Some studies have shown the down-regulation of the transcriptional levels of trehalase and trehalose phosphate synthase (tps) were observed in *Lymantria dispar* and *Apis mellifera* after sublethal exposure to different insecticides, respectively [17,18]. In addition, detoxification enzymes are known to be critical for metabolizing insecticides in invertebrates, and they can rapidly increase their activity in response to chemical stress [19]. Wang [20] observed that the expression of CYP321B1 (Cytochrome P450) in the midgut of *S. litura* could be significantly up-regulated by the sublethal concentration of betachlorpyrifos and chlorpyrifos. Moreover, Glutathione S-transferase (GST) is involved in dispelling harmful oxygen-free radicals caused by insecticide exposure [21,22], and GST has been found to be up-regulated in the resistance to phoxim and fenpropathrin in silkworm, *Bombyx mori* [23].

Metabolomics is an emerging technology that provides the real-time detection of the adaptive response of the organisms to environmental changes, pathophysiological stimuli, and genetic modifications [24,25]. So, metabolomics is a promising potential approach that could aid in understanding the stress effects of toxic pesticides on the exposed individuals. Imidacloprid induced high energy consumption, excitotoxicity, and oxidative stress (OS) in *Aphis gossypii* when exposed to a sublethal and lethal doses, but Parkinson’s disease (PD) was detected only in the lethal dose [26]. Thiacloprid could induce a reduction in serotonin activity in the honeybee brain, which was implicated in learning and behavior development [27]. Additionally, metabolomics can provide host insect metabolic pathways response during pathogenic microorganism infection. It was recently reported that glycolysis and glutaminolysis are not essential during BmNPV infection in silkworms [28]. In general, metabolomics provides a reasonable explanation for the biological functions of individuals caused by the insecticides and pathogenic microorganism stress.

In previous studies, we examined the influence of spinetoram on the detoxifying enzymes and biological characteristics in *S. frugiperda* [9,29]. To observe how metabolism and its underlying transcriptional regulation are triggered during spinetoram stress, our analysis will reveal the importance of amino acid metabolism based on a metabolome and transcriptome dataset of *S. frugiperda* larvae taken at different doses of spinetoram stress. Herein, we discovered the connections among important metabolites in physiological regulation response. It will be extremely useful to understand potential metabolic molecular mechanisms under insecticide stress on insects.

## 2. Materials and Methods

### 2.1. Insect Rearing

*S. frugiperda* larvae were obtained from a laboratory colony from the Institute of Plant Protection of the Chinese Academy of Agricultural Sciences (IPP, CAAS).

### 2.2. Test Reagent

Spinetoram (98%) was provided by Corteva Agriscience, Shanghai. The non-ionic detergent Triton X-100 and dimethyl sulfoxide (DMSO) were purchased from Beijing Excellent Biotechnology Co., Ltd. (Beijing, China). TRIzol^®^ Reagent, 1-Propanol, and 2-Propanol were purchased from Beijing Xingxian Technology Co., Ltd. (Beijing, China).

### 2.3. Acute Toxicity Assay

Diet-overlay bioassay was conducted according to the method used by Gao et al. (2020). Spinetoram was diluted with DMSO before the experiment. Formulated insecticides were diluted to generate five serial concentrations with distilled water containing 0.1% Triton X-100. Control groups were treated with 0.1% Triton X-100 only. Liquid artificial diet was dispensed into each well of a 24-well plate. After the diet cooled and solidified, 50 μL of insecticide solution was applied to the surface of each well. After the diet was dried at room temperature, one 3rd instar larva was placed in each well, and 72 larvae were tested for each concentration. Mortality was estimated after 2 days. If larvae died or could not move coordinately, they were considered dead. The LC_10_, LC_50_, and LC_90_ values for spinetoram were calculated using the Polo Plus software (v1.0, LeOra Software, Parma, MO, USA).

### 2.4. Sample Preparation

Third instar larvae of *S. frugiperda* were treated with LC_10_ and LC_90_ dose of spinetoram for 48 h with the same method as described in 2.3, before the sample collection, while a control group was fed under the same conditions with 0.1% Triton X-100 solution. Each treatment and control group comprised of ten replications that included four replications that were taken for transcriptome, and others were used for the non-targeted metabolomics. It was ensured that the content of total tissue was not less than 50 mg. All samples were stored at −80 °C for further analyses. Metabolomic and RNA-seq analysis were performed by Biomarker Technologies Co., Ltd. (Beijing, China).

### 2.5. Transcriptome Analysis

The total RNA was extracted from each sample with TRIzol method. The quality of the total RNA from individual tissue samples was evaluated with electrophoresis in 1% agarose gels, and the RNA was quantified spectrophotometrically with a NanoDrop 2000 spectrophotometer (Thermo Scientific, Waltham, MA, USA) and an Agilent Bioanalyzer 2100 (Agilent, Santa Clara, CA, USA).

Based on sequence similarity, the final assembled unigenes with length of at least 200 bp were searched using BLASTx against the following databases: Gene Ontology (GO), euKaryotic Orthologous Groups (KOG), Clusters of Orthologous Groups (COG), Swiss-Prot, Pfam, Kyoto Encyclopedia of Genes and Genomes (KEGG), evolutionary genealogy of genes: Non-supervised Orthologous Groups (eggNOG), and NCBI non-redundant nucleotide collection (nr/nt). E-value threshold of 10^−5^ was taken as the criterion of significant similarity.

### 2.6. Metabolite Extraction

An amount of 50 mg of insect sample was transferred to an Eppendorf tube. Then, the following steps were taken: Take 50 mg of one sample and homogenize it with 500 uL of ice-cold methanol/water (70%, *v*/*v*). Homogenate the mixture at 30 Hz for 2 min. After homogenization, shake the mixture for 5 min and incubate it on ice for 15 min. Centrifuge the mixture at 12,000 rpm at 4 °C for 10 min and suck supernatant 400 μL into another centrifuge tube. Add 500 μL of ethyl acetate/methanol (V, 1:3) into the original centrifuge tube, oscillate the mixture for 5 min, and incubate it on ice for 15 min. Then, centrifuge it at 12,000 rpm at 4 °C for 10 min and take 400 μL of supernatant. Merge the two supernatants and concentrate it. Then, add 100 μL of 70% methanol water into the dried product and perform ultrasonic treatment for 3 min. Finally, centrifuge it at 12,000 rpm at 4 °C for 3 min and control 60 μL of supernatant for LC-MS/MS analysis.

### 2.7. LC-MS/MS Analysis

The derivatized samples were analyzed on a Waters UPLC Xevo G2-XS QT of chromatography system coupled to an ultra-high-performance liquid chromatography–mass spectroscopy (UHPLC-MS, ExionLC, AB SCIEX; Waters, Manchester, UK). Acquity UPLC HSS T3 ((1.8 μm, 2.1 × 100 mm, Waters (Waters Technologies)) was utilized to separate the derivatives in electrospray ionization (ESI) positive (POS) and negative (NEG) ion mode. In each data acquisition cycle, dual-channel data acquisition for low and high collision energy can be carried out simultaneously. The minimal impact energy is 2 V, the high impact energy is 10–40 V, and the scanning frequency is 0.2 s. ESI-ION TRAP source parameters are as follows: Capillary voltage: 2000 V (POS) or −1500 V (NEG); Taper hole voltage: 30 V; Ion source temperature: 150 °C; Dissolvent temperature 500 °C; Reverse air flow rate: 50 L/h; De-solvent gas flow rate: 800 L/h.

### 2.8. Processing and Statistical Analysis of Metabolomics Data

The original data file obtained by LC-MS analysis is firstly converted into mzML format by ProteoWizard software. Peak extraction, alignment, and retention time correction are performed by XCMS program. The “SVR” method was used to correct the peak area. Peaks with a deletion rate > 50% in each group of samples were filtered. After that, metabolic identification information was obtained by searching the laboratory’s self-built database and integrating the public database and mtDNA. Finally, statistical analysis was carried out by the R program. Statistical analysis includes univariate analysis and multivariate analysis. Univariate statistical analysis includes Student’s *t*-test and variance multiple analysis. Multivariate statistical analysis includes principal component analysis (PCA), partial least squares discriminant analysis (PLS-DA), and orthogonal partial least squares discriminant analysis (OPLS-DA).

## 3. Results

### 3.1. Toxicity of Spinetoram to S. Frugiperda 3rd Instar Larvae

The toxicity of spinetoram to 3rd larvae of *S. frugiperda* was determined by the method of diet-overlay bioassay. The LC_10_ value was estimated as 0.172 mg/L (95% limit: 0.125 and 0.214 mg/L, respectively), and the LC_90_ value was estimated as 0.725 mg/L (95% limit: 0.648 and 0.836 mg/L, respectively). The LC_10_ and LC_90_ values were used as the stress doses on *S. frugiperda* for transcriptome and non-targeted metabolomics experiments, respectively.

### 3.2. Transcriptome Analysis of S. Frugiperda in Responses to Spinetoram Application

For transcriptome profiling, three treatments: Control (CK), Spin-10 (LC_10_), and Spin-90 (LC_90_), in four replicates at 48 h post *S. frugiperda* stress were used to construct 12 cDNA libraries. A total of 83.29 Gb clean data was obtained and the Q30 base percentage was 93.03% and above. Clean reads were aligned with a designated reference genome (assembly ZJU_Sfru_1.0) and the comparison efficiency ranged from 82.35 to 84.17% (Table 1). To evaluate biological replicate correlation in control and different doses, we analyzed Pearson’s correlation coefficient analysis among all treatments (Figure 1A). The abundance of down-regulated genes was higher than up-regulated genes as compared to control at LC_10_ dose (Figure 1B), among them, 1627 unigenes were up-regulated and the remaining 1831 unigenes were down-regulated. However, the abundance of genes was reversed compared to control at LC_90_ dose (Figure 1B,C), among them, 1804 unigenes were up-regulated and the remaining 1580 unigenes were down-regulated. The color difference indicates high (red) and low (green) expressions.

To explore the response of different processes in control and toxic treatments, we analyzed the KEGG enrichment analysis of common differentially expressed genes (DEGs). The sets of DEGs were assigned to significant KEGG pathways (*p* > 0.05), which in Control-LC_10_, Control-LC_90,_ and LC_10_-LC_90_ at 48 h, were listed in Figure 1D. The KEGG pathways were analyzed for genes enriched in a higher number of metabolic processes in all treatments (Figure 1F) and genes enriched in a higher level of cytochrome P450 in LC_10_ vs. LC_90_. Moreover, we observed a range of important metabolic pathway genes and detoxification enzyme genes after spinetoram stress, for example, 5-hydroxytryptamine receptor, acetylcholinesterase, catalase, connectin, superoxide dismutase, cytochrome P450, glutamate synthase, etc. (Table 2).

Both positive and negative ion modes of ESI were used during LC-MS/MS analysis. A comprehensive and extensive analysis of metabolic profiles from LC_10_, LC_90,_ and control groups were performed by multiple orthogonal methods. Firstly, the principal component analysis (PCA) was performed to analyze the metabolic profiles. The PCA showed a no-complete separation among the LC_10_, LC_90_, and control groups at pos modes (Figure 2A). The principal component analysis (PCA) model was performed to analyze the metabolism profiles. The PCA model showed a rough separation among the LC_10_, LC_90_, and control groups at the pro-model. The PCA models obtained from LC_10_-control and LC_90_-control showed a complete separation, indicating a significant difference in metabolic characteristics. However, no-complete separations were obtained from LC_10_-control and LC_90_-control at neg-modes (Figure 2B).

Using statistics based on metabolite identity determined by analysis of both POS and NEG mode, a total of 715,501 metabolites in *S. frugiperda* were significantly changed after spinetoram infection, respectively. A total of 66 metabolites were significantly up-regulated, and 225 metabolites were significantly down-regulated at LC_10_-pos and LC_90_-pos (Figure 3A). In the other model, a total of 114 and 161 metabolites were significantly up-regulated at LC_10_-neg and LC_90_-neg while 94 and 132 metabolites were significantly down-regulated at these time points (Figure 3A). To further observe patterns of overall metabolite abundance, the heat map analysis was performed, which displayed a significant difference in the abundance of metabolites between control group, LC_10_, and LC_90_. In addition, we have further identified metabolites that associated with spinetoram treatment (Figure 3B,C). After 48 h of stress, most metabolites were significantly down-regulated compared to the control samples via pos-mode (Figure 3A–C). However, most metabolites were significantly up-regulated compared to the control samples via neg-mode (Figure 3A,D,E).

Metabolic pathway disruption was further investigated in *S. frugiperda* exposed to LC_10_ and LC_90_ of spinetoram using MetaboAnalyst, based on metabolites (Figure 4). The most significantly modified pathways between LC_10_ and the control group were the biosynthesis of amino acids, carbon metabolism, ubiquinone and other terpenoid-quinone biosynthesis and glycine, serine, and threonine metabolism (Figure 4A). The changes in cysteine and methionine metabolism, arachidonic acid metabolism, glycerophospholipid metabolism, linoleic acid metabolism, and terpenoid backbone biosynthesis were characterized in the LC_90_-control group at neg-mode (Figure 4B). Furthermore, the most significant changes were observed in ABC transports, biosynthesis of amino acids, bile secretion, arginine and proline metabolism, phenylalanine metabolism, pantothenate, and CoA biosynthesis in pos-mode (Figure 4C,D). In contrast with the observation that metabolites were mainly enriched in amino acid metabolism pathways, metabolites at neg-mode were enriched in amio sugar and nucleotide sugar metabolism between LC_10_ and LC_90_ groups (Figure 4E,F). Moreover, the significant metabolic pathways were summarized to reveal the effects of the spinetoram in *S. frugiperda*, based on biomarkers (Table 3).

## 4. Discussion

Metabonomic has been playing an important role in understanding metabolic information [30]. In recent years, it has been widely used in the study of diverse areas, including diseases, plants, microbiology, nutrition, toxicology, and insects [31]. During our study, we focused on the toxicity of pesticides of fall armyworm larvae following spinetoram treatment and identified 715 and 501 metabolites with differential abundance at LC_10_ and LC_90_ in different models, respectively. Among all differential metabolites, including diverse amino acids, other compounds were confirmed and were found to be involved in carbon metabolism, serine and threonine metabolism, glycerophospholipid metabolism, linoleic acid metabolism, ABC transports, pantothenate, and CoA biosynthesis. Carbohydrate metabolisms were extremely varied in *S. frugiperda* exposed to a low dose. Glucose and lactic acid both play important roles in glycolysis and glucogenesis as part of the starch and sucrose metabolic pathways [32]. In our study, glucose was up-regulated in spinetoram studies relative to the control group, indicating an increase in energetics necessary for detoxification following pesticide exposure. Similar phenomena were observed in other insects in which supply of glucose was up-regulated after exposure to neonicotinoid to supply energy for detoxification of pesticides [27].

Amino acids are biologically important molecules that could also serve as an energy source [33]. In the treatment group, the metabolism of valine and isoleucine were down-regulated to reduce the accumulation of acetyl-CoA. The metabolism of leucine was up-regulated to make up for deficiency in the accumulation of acetyl-CoA and maintain the conservation of citric acid. Phenylalanine metabolism could produce tyrosine, and fumaric acid was the end product of tyrosine [34,35]. So, the down-regulation of phenylalanine may reduce the amount of fumaric acid, and then retard the tricarboxylic (TCA) cycle to produce GTP under spinetoram stress. In addition, phenylalanine are essential amino acids for mammals; they can be synthesized by the gut microbiota [36,37,38]. The LC_90_ treatment of spinetoram may affect the activity of advocate microorganisms and reduce the efficiency of phenylalanine synthesis, which may be the reason for the decrease in the abundance of phenylalanine under LC_90_ treatment. In animals, lysine can also induce apoptosis in neuronal cells [39]. The abundance of lysine was decreased after spinetoram exposure, which damages the nervous system of the larvae. The arginine could be converted into phosphoarginine by using arginine kinase (AK) [40]. The abundance concentration of arginine was increased under spinetoram stress, which may positively impact energy utilization and muscle contraction. The arginine was increased causing non-functional contraction of the insect’s muscles, stimulating the larva to spend excess energy. Additionally, pipecolic acid is a major metabolite of lysine and may facilitate GABAerguc transmissions by stimulating GABA release, inhibiting GABA uptake, or enhancing GABA_A_ receptor responses [41,42,43,44]. The reduction of lysine in the LC_90_ group indicated pipecolic acid was influenced, which is essential for the transmission of neural substrates in *S. frugiperda* larvae. Amino acid variation was one of the important factors that causes the death of *S. frugiperda* larvae exposed to spinetoram.

Glutamine is utilized in the cell as an essential source of carbon for the synthesis of macromolecules and the production of energy [45]. Glutamine was decreased in the LC_90_ group, indicating that *S. frugiperda* reduced production of energy. Glutamine can be hydrolyzed by glutaminase to glutamate and ammonia and is an important source of glutamate. However, the decrease in glutamine did not affect the decrease in glutamate abundance in *S. frugiperda* after exposure to spinetoram. On the contrary, the abundance of glutamate was increased in the LC_90_ group when exposed to spinetoram. Further, exposure to spinetoram also led to increased glutamate levels, which is also associated with increased intestinal energy requirements [46]. Glutamate serves as one of the major excitatory neurotransmitters in the central nervous system, which induces excitotoxicity in individuals when exposed to neurotoxicants and forms a glutamate-gated chloride channel (GluCls) with ionic inhibitory glutamate receptor (IGuRs) [47,48]. The nAchRs, by binding with nicotine, can stimulate the release of glutamate [49]; this may be a cause of the abundance of glutamate being increased after exposure to spinetoram. Aspartic acid also performed as an excitatory neurotransmitter in the central nervous system [50], so the abundant concentration of aspartic acid could lead to excitotoxicity in the spinetoram treatment. Therefore, the glutamine, glutamate, and aspartic metabolites could be potentially used as biomarkers of excitotoxicity in *S. frugiperda* in response to the stress of spinetoram. Excitotoxicity could be one reason for the high mortality of *S. frugiperda* in the spinetoram exposure.

Reactive oxygen species (ROS) could be formed after the application of the insecticide, which resulted in DNA damage and cell apoptosis in exposed individuals [51]. When the organism is in a certain pathological state (e.g., nicotinic toxins), the oxidative capacity in the body exceeds the antioxidant capacity and causes oxidative stress [52,53]. Some researchers indicated that the excessive production of ATP and the glutamate excitotoxicity were induced by bioenergetic disruption, which stimulates the formation of ROS [54,55]. So, increased levels of glutamate are another evidence that the larvae were being affected by an oxidative stress response. The high energy consumption and excitotoxicity elevated the ROS in *S. frugiperda* when exposed to LC_10_ and LC_90_ concentrations of the spinetoram. Most organisms have evolved antioxidant defense mechanisms to maintain cellular redox homeostasis (e.g., antioxidant enzymes: catalase (CAT), superoxide dismutase (SOD), glutathione reductase (GR), glutathione S-transferase (GST), and glutathion (GSH)), which are significantly affected by pesticides exposure [56,57]. Meanwhile, ascorbate (ASA) is the primary cellular antioxidant and is altered the transport and metabolic mechanisms responsible for dehydroascorbic acid (DHA) recycling by oxidative stress [58,59]. The ASA-GSH cycle, in which ASA participates, is considered to be an essential system for scavenging reactive oxygen species under environmental stress [60]. The up-regulation of ASA in the LC_90_ group further indicates the increased ability in scavenging of ROS from oxidative pressure in *S. frugiperda*. Vitamin E (alpha-, beta-, gamma-, and delta-tocopherol) also serves as one of the major antioxidants in organisms, whereas antioxidant activity tocopherol was decreased by the addition of (−)-epicatechin (−)-epigallocatechin gallate [61]. We found that the down-regulation of delta-tocopherol was influenced by the up-regulation of epigallocatechin gallate in the LC_90_ group. Arachidonic acid, per se, can help alleviate oxidative stress [62], and the up-regulation of arachidonic acid has a positive impact on scavenging ROS in spinetoram stress. Moreover, under the spinetoram, the increased arachidonic acid can induce inflammation, which may be a further reason for larvae death. Nicotinamide adenine dinucleotide (NAD) and its phosphorylated form, NADP+, are the major coenzymes of redox reactions in central metabolic pathways. Besides its crucial role in cellular metabolism, NAD also serves as the substrate of several families of regulatory proteins: protein deacetylases (sirtuins), ADP-ribosyltransferases (ARTs), and poly (ADP-ribose) polymerases (PARPs), which govern vital processes including gene expression, DNA repair, apoptosis, aging, cell cycle progression, and many others [63,64,65]. We found that the NAD and NADP+ were up-regulated at LC_90_ group, compared to the control group. An increase in the levels of NAD and NADP+ confirms the induction of oxidative stress on exposure with these pesticides. In addition, it has been shown that NADH and NADPH are able to directly scavenge oxidants such as hypohalous acids produced during inflammation [66]; they play an active role in the larval removal of LC_90_ spinetoram on themselves. However, they did not change at low dose concentration, and SOD may play a role in the larval antioxidant response (Table 2).

Tryptophan is an essential amino acid for protein synthesis, and some important physiological metabolism, and participates in immune function [67]. 5-hydroxytryptamine (5-HT), which is synthesized by tryptophan, plays an important role in central neurotransmitters [68,69]. Degradation of tryptophan, resulting in lower blood levels and impaired cerebral production and release of serotonin, is enhanced by inter alia inflammation, pregnancy, and stress in all species investigated, including humans [70]. The spinetoram-induced ROS resulted in the decrease in tryptophan levels in larvae and the decrease in 5-HT levels. However, we found 5-HT were up-regulated, compared with the control group, in spinetoram stress and have a positive impact on scavenging ROS from oxidative pressure. We inferred that the decrease in tryptophan was attributed to increased NAD, NADP, and 5-HT synthesis in insects to counter oxidative stress in larvae. Immune responses and metabolic regulation are tightly coupled in all animals including insects, but the underlying mechanistic connections are far from clear [71,72]. 5-HT is mainly found in the mucous membranes of the digestive tract and is stored in small amounts in the central nervous system. It must be pointed out that only a minority of the metabolites in the hemolymph may have originated from hemocytes and that most metabolites that enter the circulatory system are produced by other tissues, mainly the fat body [28]. Pyrethroids inhibit KYNA synthesis and tryptophan metabolism, leading to de novo generation of nicotinamide adenine dinucleotide coenzyme (NAD+) [73], and may be one of the reasons for the increase in NAD under LC_90_ treatment. Therefore, we should take into account the changes in metabolites affected by toxic substances or pathogenic microorganisms on different tissues or sites, which will be more conducive to our understanding of insect metabolic responses.

That nAChRs are widely expressed in the central nervous system and numerous functions of acetylcholine signals, have been elucidated, including the induction of dopamine (DA) [74,75]. Sporadic or idiopathic PD is a neurodegenerative disease, and the hallmark symptoms of PD are related to DA deficiency [76,77]. DA, one of the monoamines, is a neurotransmitter synthesized from tyrosine. The decrease in tyrosine was detected in the LC_90_ group, indicating that the pathogenesis of the PD appeared in insects by exposure to a lethal dose of spinetoram, and it may further explain the high mortality in *S. frugiperda* after exposure to a lethal dose of spinetoram. Chlorfenapyr-treated zebrafish brain metabolome revealed a significant decrease in the metabolites such as tyrosine in high concentration [78] and is consistent with our experimental results. Additionally, sequential injections with paraquat resulted in a significant loss of dopaminergic neurons in the *substantia nigra* (SN) in a previous study [79]. Some scholars believed that the cause of PD is the loss of DA neurons and the subsequent imbalance of the DA/acetylcholine concentration ratio [80]. The pathogenesis of PD can depend on the self-regulation of the central nervous system to dynamically reshape the balance of DA/acetylcholine. So, we observed the decrease in acetylcholine in the LC_90_ group compared with the LC_10_ and control groups. We inferred that the appearance of this phenomenon also provides the conditions for the larvae to return to normal activities.

## 5. Conclusions

Physiological and biochemical changes in *S. frugiperdai* under the sublethal and lethal stress of spinetoram were analyzed based on transcriptome and LC-MS/MS analysis by identifying the regulated metabolites. Comprehensively, abnormal metabolism of amino acids causes energy consumption, excitotoxicity, ROS, and PD, which were the main cause of larval death under spinetoram exposure. The accumulations of lysine indicated the non-functional contraction of the insect’s muscles and caused the larvae to spend excess energy. The accumulations of glutamate and asparagine caused the excitotoxicity in *S. frugiperda* in the LC_90_ group. The increase in the biosynthesis of antioxidants demonstrated the response of *S. frugiperda* scavenged ROS under spinetoram exposure. The decrease in PD in the LC_90_ group suggested that spinetoram exposure was an external cause for the occurrence of PD. In general, our findings provide new insights into the changes of amino acids in larvae under pesticide exposure.

## Figures and Tables

**Figure 1 insects-13-00852-f001:**
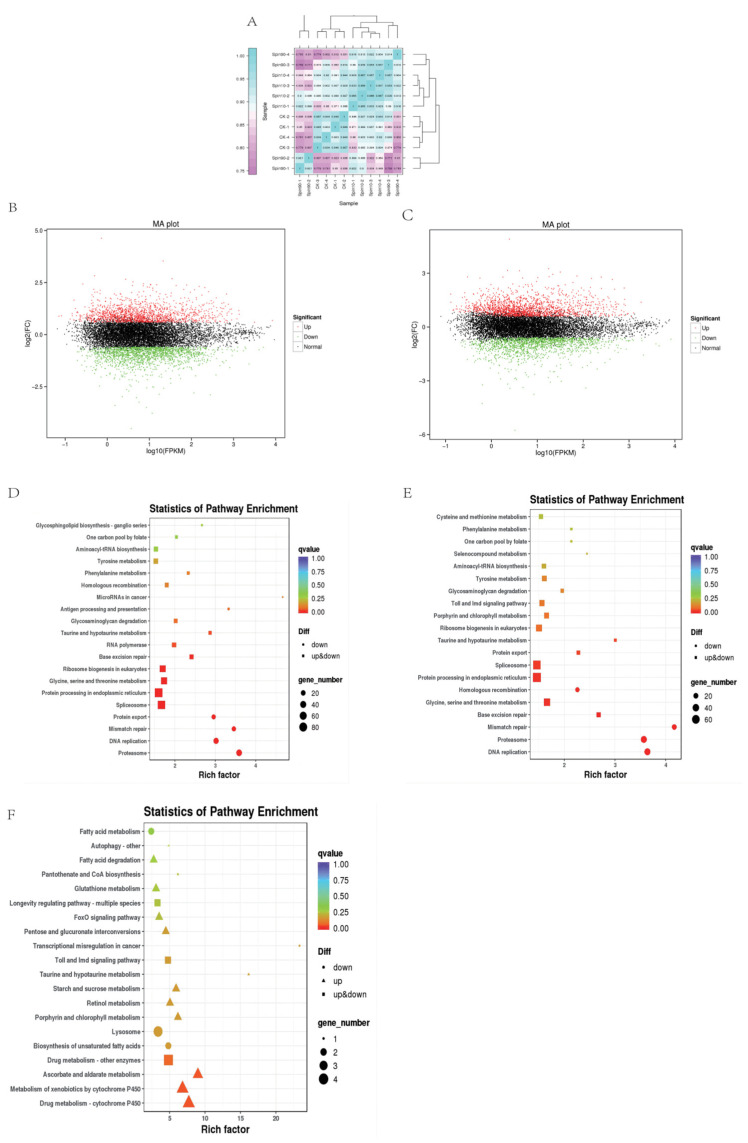
Transcriptome response of *S. frugiperda* to spinetoram stress. (**A**) Pearson’s correlation coefficient analysis. (**B**,**C**) Derived from volcano plots of transcriptome in Control vs. LC_10_ and Control vs. LC_90_, respectively. KEGG pathways in Control vs. LC_10_ (**D**), Control vs. LC_90_ (**E**), and LC_10_ vs. LC_90_ (**F**). Metabolic profiles of *S. frugiperda* in response to spinetoram treatment.

**Figure 2 insects-13-00852-f002:**
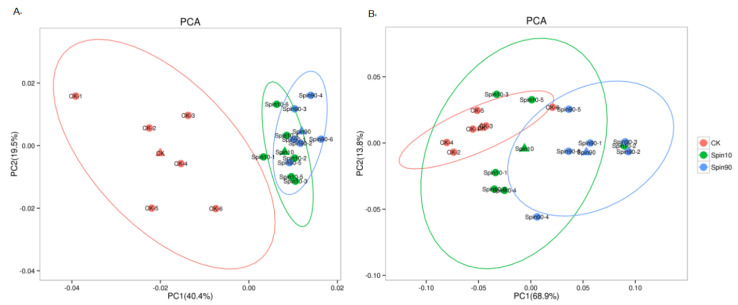
Metabolic profile in *S. frugiperda* exposed to sublethal and lethal dose of the spinetoram. (**A**) Principal component analysis (PCA) model of Spin10, Spin 90, and Control at pro-model, (**B**) partial least squares-discriminant analysis (PLS-DA) model of Spin10, Spin90, and Control at neg-model.

**Figure 3 insects-13-00852-f003:**
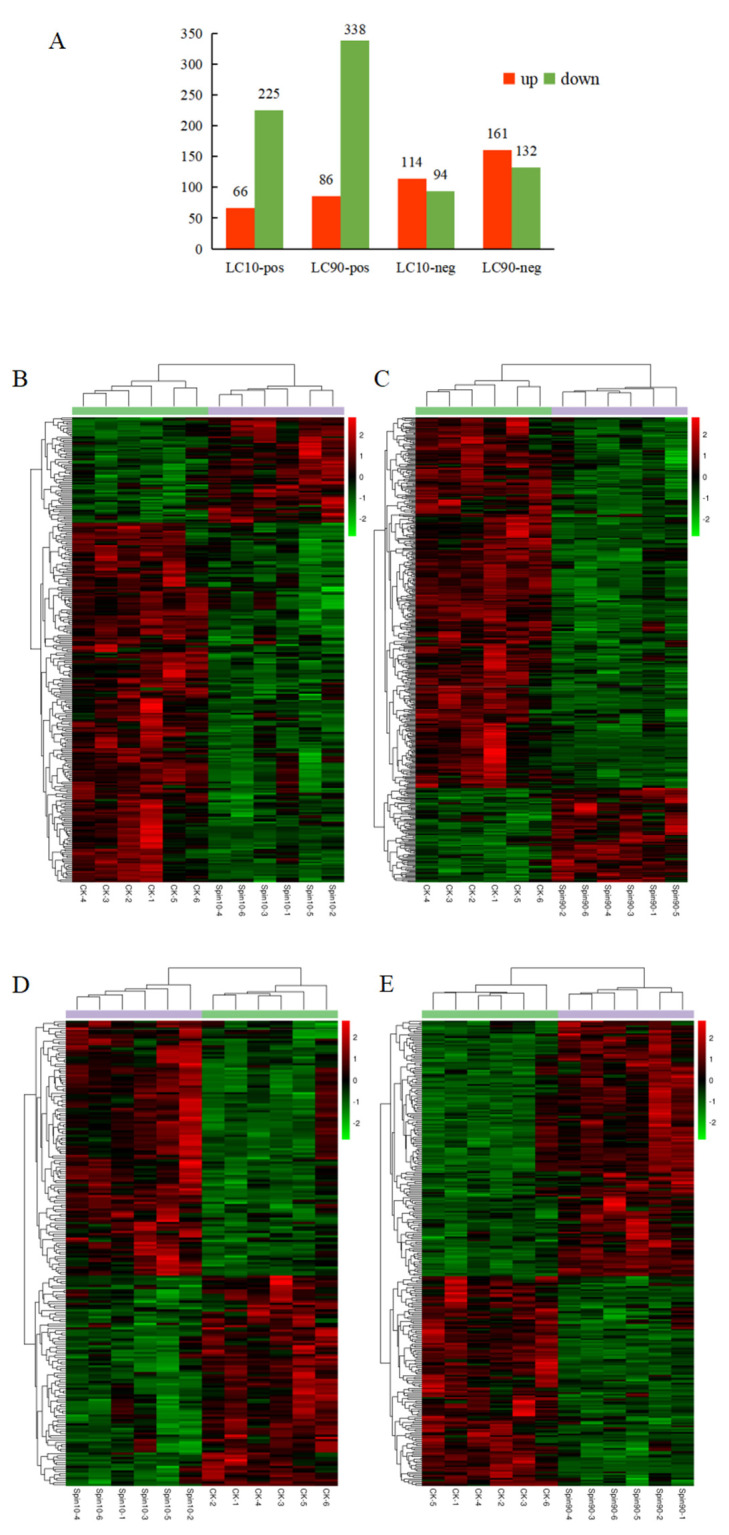
Analysis of differentially expressed (DE) metabolites in *S. frugiperda* to spinetoram stress. (**A**) Numbers of metabolites that were up-regulated (red) and down-regulated (green) in spinetoram-stressed fall armyworm larvae. (**B**–**E**) Heatmaps of DE metabolites in fall armyworm at pos-mode (**B**,**C**) and neg-mode (**D**,**E**) after spinetoram stress. Each column represents one sample, and each row represents one DE metabolite. Red color represents the relative level of the up-regulated metabolites, and green color represents down-regulation Metabolic pathway analysis.

**Figure 4 insects-13-00852-f004:**
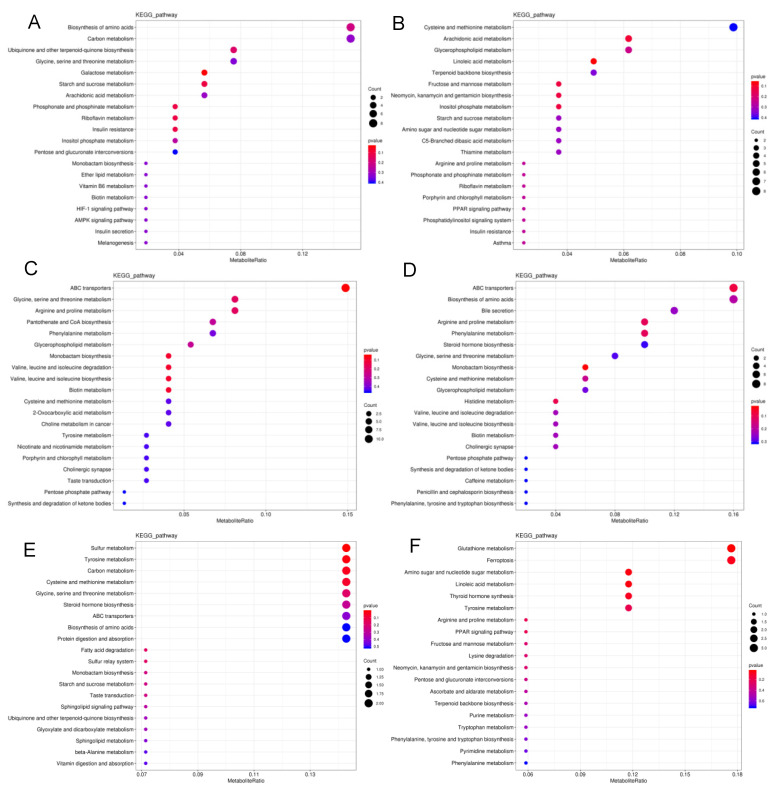
Summary of pathway analysis. (**A**) Pathway analysis of LC_10_ and control groups at neg-mode. (**B**) Pathway analysis of LC_90_ and control groups at neg-mode. (**C**) Pathway analysis of LC_10_ and control groups at pos-mode. (**D**) Pathway analysis of LC_90_ and control groups at pos-mode. (**E**) Pathway analysis of LC_10_ and LC_90_ groups at neg-mode. (**F**) Pathway analysis of LC_10_ and LC_90_ groups at pos-mode. The color and size of the shapes represent the regulated degree of the pathways by the treatment of doses of the spinetoram; larger, red shapes indicate a greater effect on the pathway.

**Table 1 insects-13-00852-t001:** Summary of transcriptome sequencing data.

	Sample	Clean Reads	Clean Bases	Mapped Read	Guanine-Cytosine Content (%)	Q20(%)	Q30(%)
CK	Control-1	24,804,561	7,399,530,118	36,072,981 (72.71%)	48.04	98.29	94.93
	Control-2	25,406,000	7,581,035,800	367,63,311 (72.35%)	48.00	97.86	94.93
	Control-3	20,529,036	6,132,802,190	29,967,481 (72.99%)	47.78	97.94	93.77
	Control-4	21,455,769	6,405,367,636	31,161,221 (72.62%)	49.17	97.71	93.29
LC_10_	Spin10-1	24,388,290	7,285,860,770	34,931,682 (71.62%)	48.22	97.95	93.82
	Spin10-2	23,612,040	7,059,990,216	33,693,849 (71.35%)	47.86	97.71	93.32
	Spin10-3	23,612,040	6,567,603,744	31,630,174 (71.99%)	47.92	97.59	93.03
	Spin10-4	21,780,281	6,504,361,012	31,270,395 (71.79%)	48.20	97.78	93.44
LC_90_	Spin90-1	20,923,896	6,261,061,732	29,946,614 (71.56%)	47.73	97.63	93.03
	Spin90-2	20,796,109	6,220,125,194	29,921,237 (71.94%)	47.40	97.64	93.10
	Spin90-3	25,473,623	7,599,203,900	36,194,890 (71.04%)	48.25	97.83	93.53
	Spin90-4	27,731,793	8,274,619,114	39,174,512 (70.63%)	48.49	97.86	93.65

**Table 2 insects-13-00852-t002:** Changes of key genes in *S. frugiperda* by exposure to LC_10_ and LC_90_ concentration of the spinetoram compared to the control group (*t*-test *p* < 0.05).

Unigene ID	Gene Annotation	Type of Regulation	
		LC_10_ Group	LC_90_ Group
LOC118266960	5-hydroxytryptamine receptor	Up	Up
LOC118275809	5-hydroxytryptamine receptor 1A	-	Up
LOC118262451	acetylcholinesterase	Up	Up
LOC118268389	acetylcholine receptor subunit alpha	Up	Up
LOC118275478	cholinesterase 2	-	Up
LOC118276148	esterase B1	-	Up
LOC118276231	juvenile hormone esterase-like	Down	Down
LOC118274652	glucose dehydrogenase	Up	Up
LOC118277268	glucose-1-phosphatase-like	Up	-
LOC118277428	facilitated trehalose transporter Tret	Up	-
LOC118277941	UTP--glucose-1-phosphate uridylyl transferase	Up	-
LOC118269585	glucose transporter type	Up	Up
LOC118280239	dehydrogenase	Up	Up
LOC118280972	UDP-N-acetyl hexosamine pyrophosphorylase-1	Up	Up
LOC118262085	acetyl-coenzyme A synthetase	-	Down
LOC118266444	3-ketoacyl-CoA thiolase	Up	-
LOC118280410	acetyl-CoA carboxylase	Up	Up
LOC118267788	connectin	-	Up
LOC118266022	hydroxymethylglutaryl-CoA synthase	-	Down
LOC118281649	2-oxoglutarate dehydrogenase	-	Up
LOC118278502	ATP-citrate synthase	Up	-
LOC118262464	isocitrate dehydrogenase [NADP] cytoplasmic	Down	Down
LOC118266957	4-aminobutyrate aminotransferase	Up	Up
LOC118272711	gamma-aminobutyric acid receptor	Up	Up
LOC118262156	glutamate synthase [NADH]	Up	Up
LOC118265687	glutamyl-tRNA (Gln) amidotransferase subunit B	Up	Up
LOC118282360	asparagine synthetase	Down	Down
LOC118268389	acetylcholine receptor subunit alpha	Up	Up
LOC118276145	muscarinic acetylcholine receptor DM1	Up	Up
LOC118282418	catalase	Down	Down
LOC118266989	superoxide dismutase	Up	Up
LOC118269785	glutathione S-transferase 1	Down	Down
LOC118271638	glutathione S-transferase 2	Down	Down
LOC118268807	glutathione hydrolase 1 proenzyme	Up	Up
LOC118280652	carbonyl reductase [NADPH] 3	Down	Down
LOC118266432	leukotriene A-4 hydrolase	-	Down
LOC118269281	glutaryl-CoA dehydrogenase	Up	Up
LOC118273421	tyrosine aminotransferase	Up	Up
LOC118267609	dopamine N-acetyltransferase	-	Up
LOC118277744	venom carboxylesterase-6	Up	Up
LOC118281189	cytochrome P450 307a	Down	Down
LOC118274175	cytochrome P450 6B6	Down	Down
LOC118270797	cytochrome P450 4C1	Up	Up
LOC118270458	cytochrome P450 4c21	Up	Up
LOC118272240	cytochrome P450 4c3	Up	Up
LOC118282431	cytochrome P450 4d2	Up	Up
LOC118266763	cytochrome P450 6a2	Up	Up
LOC118273800	cytochrome P450 6B2	Up	Up
LOC118262785	cytochrome P450 6B5	Up	Up
LOC118264635	cytochrome P450 9e2	Up	Up
LOC118271250	cytochrome P450 301a1	Up	Up
LOC118279412	UDP-glucuronosyltransferase 2B15	Up	Up
LOC118269333	UDP-glucuronosyltransferase 2B19	Up	Up
LOC118279357	UDP-glucuronosyltransferase 2B7	Up	Up

**Table 3 insects-13-00852-t003:** Changes of key metabolites in *S. frugiperda* by exposure to LC_10_ and LC_90_ dose of the spinetoram compared to the control group (*t*-test *p* < 0.05).

Metablite ID	Metabolite Name	LC_10_ Group	LC_90_ Group
neg_127	D-Glucose 6-phosphate	Up	Up
neg_149	Dehydroascorbic acid	-	Up
neg_39	Tyrosine	-	Down
neg_399	Arginine	Up	Up
neg_412	Leucine	Up	Up
neg_58	NADP+	-	Up
neg_76	Glutamate	Up	Up
pos_136	L-Valine	Down	Down
pos_194	Aspartic acid	Up	Up
pos_198	Tryptophan	-	Up
pos_379	Arachidyl carnitine	-	Up
pos_517	delta. -Tocopherol	-	Up
pos_613	DL-Phenylalanine	-	Down
pos_732	Glutethimide	-	Down
pos_752	Acetylcholine	-	Down
pos_766	Allysine	Down	Down
pos_79	NAD	-	UP
pos_904	Epigallocatechin gallate	-	Down
pos_99	L-Isoleucine	-	Down

## Data Availability

The data presented in this study are available on request from the corresponding author. The data are not publicly available due to restriction e.g., privacy.

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
