# Peer review of "Transcriptome and Metabolome Analysis Reveals the Importance of Amino-Acid Metabolism in Spodoptera Frugiperda Exposed to Spinetoram"

_insects, 2022, doi:10.3390/insects13090852_

Round 1

Reviewer 1 Report

Comments to the Author

This study investigated the effect of spinetoram on amino-acid metabolism of fall armyworm, Spodoptera frugiperda by using metabolome analysis. And transcriptome and LC-MS/MS analysis revealed the effects of pesticide exposure on amino-acid metabolism of larvae. However, the evidences are not adequate to support the conclusion, see my concerns below. The language is also needed to be improved. Several questions or suggestions are appended below.

1. The resolution of Figure 2 and Figure 4 is too low, and the text on the left is not clear enough. Can the author provide a higher resolution picture?

2. The authors identified a number of differentially expressed genes in S. frugiperda by exposing to LC10 and LC90 dose of the spinetoram compared to the control. Can the authors select some key genes for qRT-PCR validation to ensure that the data will be more enriched.

3. In Table 3, several key metabolites are up-regulated or down-regulated in LC90 group compared with the control group. However, these key metabolites did not change significantly in the LC10 group. Can the author give a reasonable explanation and discussion?

Author Response

  1. The resolution of Figure 2 and Figure 4 is too low, and the text on the left is not clear enough. Can the author provide a higher resolution picture? 

I have provided the journal with high-definition images

  1. The authors identified a number of differentially expressed genes in S. frugiperda by exposing to LC10 and LC90 dose of the spinetoram compared to the control. Can the authors select some key genes for qRT-PCR validation to ensure that the data will be more enriched.

 i do not have assess to lab and samples right now so this may be little difficult to perform.

  1. In Table 3, several key metabolites are up-regulated or down-regulated in LC90 group compared with the control group. However, these key metabolites did not change significantly in the LC10 group. Can the author give a reasonable explanation and discussion?

I have provided additional discussion and explanation for this section

Reviewer 2 Report

Dear Authors,              

I carefully read the submitted manuscript discussing transcriptomic and metabolome analysis of amino-acid metabolism in Spodoptera frugiperda exposed to Spinetoram. The topic is interesting and up-to-date as more studies are focusing on sublethal and lethal effects of pesticides/insecticides nowadays. The sublethal effects are even important as they may lead to explanations related to development of resistance. In Introduction I miss mentioning of this phenomenon which is of utmost importance. I miss a clear hypothesis formulation why the study from this aspect is necessary and what is expected from the outcome. Please also explain why the two extreme concentrations were chosen as it is obvious that very likely significant differences will be found in the surviving individual’s tissues. Why for example LC30 or LC50 was not studied or communicated as better representatives of sublethal concentrations.

In general I have placed anonym sticky notes on the PDF for suggested changes, remarks, questions. During corrections please consult those. As for references I highlighted where corrections are to be made (mainly species names to be in italic).

In Materials and Methods the Metabolite Extraction description is fully confusing and unclear especially between lines 162-169. Please rewrite it and explain exactly what was subjected to LC-MS/MS analysis.

In Discussion please better highlight the finding regarding amino-acid metabolism following Spinetoram treatment.

As for Figures, Tables and related legends I also made my comments on the notes.

I suggest minor revision before acceptance.

Kind regards,

Reviewer

Author Response

1.In general I have placed anonym sticky notes on the PDF for suggested changes, remarks, questions. During corrections please consult those. As for references I highlighted where corrections are to be made (mainly species names to be in italic).

I have read the relevant content and revised it according to your comments

  1. In Materials and Methods the Metabolite Extraction description is fully confusing and unclear especially between lines 162-169. Please rewrite it and explain exactly what was subjected to LC-MS/MS analysis.

I rewrote the section for accuracy

  1. In Discussion please better highlight the finding regarding amino-acid metabolism following Spinetoram treatment.

I have provided additional discussion and explanation for this section

  1. As for Figures, Tables and related legends I also made my comments on the notes.

It has been revised according to the indirection you provided

Round 2

Reviewer 1 Report

The authors have made major revisions and enhancements to the manuscript, which may be considered for acceptance and publication in the current form.